# The Cost-Effectiveness Analysis of the Productivity Measurement and Enhancement System Intervention to Reduce Employee Work-Related Stress and Enhance Work Performance

**DOI:** 10.3390/ijerph19042431

**Published:** 2022-02-19

**Authors:** Irene Jensen, Zana Arapovic-Johansson, Emmanuel Aboagye

**Affiliations:** Institute of Environmental Medicine, Karolinska Institute, 171 77 Stockholm, Sweden; irene.jensen@ki.se (I.J.); bozana.johansson@ki.se (Z.A.-J.)

**Keywords:** organizational intervention, work-related stress, objective workload, average cost-effectiveness ratio, interrupted time series

## Abstract

Background: The study evaluates the cost-effectiveness of the Productivity Measurement and Enhancement System (ProMes) intervention to reduce employee work-related stress and enhance work performance. Methods: A prospective cohort study was used to undertake the evaluation from a business perspective. Objective workload data and stress were gathered repeatedly over a 17-month period (i.e., before and after intervention). Independent *t*-test and an interrupted time series (ITS) analysis were used in the analysis. The average cost-effectiveness ratio (ACER) was calculated as a ratio of the average cost of the intervention and the effect sizes of the different outcomes to reflect the average cost per clinician for each unit change in outcome. Results: Based on the results of the ITS analysis, an expenditure of EUR 41,487 was linked with no change in stress levels, according to the ACER for stress. In addition, the expenditures associated with each unit change were EUR 3319 for overall tasks per hour worked, EUR 2761 for visits per hour worked, EUR 2880 for administrative tasks, but EUR 9123 for answering phone calls. Conclusions: ProMes is not cost–effective in terms of work-related stress levels, but the intervention seemed to have increased efficiency in some objective work performance measures, albeit at a relatively high extra cost.

## 1. Introduction

Work-related stress is prevalent and a major challenging issue facing occupational safety and health internationally [1,2,3]. After low back pain, it is the second most frequent cause of sickness absence in many European countries, such as the UK and Finland. In Sweden, it is the most frequent cause of sickness absence [4]. Stress-related diagnoses, together with mood and anxiety diagnoses, are associated with longer sick leave than most other illnesses in Sweden [2,5]. According to the European Survey of Enterprises on New and Emerging Risks [6], 79% of European managers are concerned and find it difficult to manage work-related stress in their workplaces. Further, only a small number of organizations in Europe have systematic procedures to deal with workplace stress [6].

The definition of work stress and its measurement adopt many different approaches, including the engineering approach (i.e., load and demand), physiological approach (non-specific changes in the biology of an individual), and the psychological approach (person–environment interaction) [1,7]. The more contemporary theories of stress that encompass the person’s interaction with their work environment are also concerned with the psychological mechanisms underpinning that interaction [1,7]. Thus, the individual’s subjective assessment can be a valid measure of work stress according to the psychological approach of defining stress. Some of the challenges with the definition and measurement of work stress also raise an important question of the determination of carry-over effects—positive or negative. The focus on work stress may lead one to suppose that work has solely a negative impact on health; however, it is not the case. There is an indication that, in some cases, working can be beneficial to one’s health through improving psychological well-being [7,8].

While low levels of work stress can be motivating, higher levels can impede individual performance and organizational productivity. The research on the stress, health and performance connection shows that work-related stress can directly cause absence from work and staff turnover, along with decreased performance and productivity [9]. Most studies on health and productivity have addressed changes in productivity and associated costs to employers that resulted from declining employee health [10]. Nevertheless, methods to measure and estimate the individual and organizational level of performance have not been straightforward. Self-report productivity assessments are a standard part of many organizations’ employee evaluation systems. The primary functions of self-reported productivity measures are to determine the extent to which health status influences performance (descriptive), to assess the differential impact of various health risks on performance (comparative), and to assess change over time, particularly as part of program evaluation (evaluative) [11]. There are few organizations that use objective measurement, and most organizations, such as those in the information and service sectors, cannot quantify individual performance without sacrificing certain qualitative attributes.

Work-related stress can have a significant impact on the health of individuals, as well as an impact on organizational and national economies. In Europe, the annual costs of absenteeism and lost productivity while at work (i.e., presenteeism) because of stress and depression are estimated at € 72 billion [9,12]. In the EU, the cost of work-related depression was estimated to be € 617 billion annually at the societal level [12]. The total estimated included costs to employers resulting from absenteeism and presenteeism (€ 272 billion), loss of productivity (€ 242 billion), healthcare costs of € 63 billion, and social welfare costs in the form of disability benefit payments (€ 39 billion). In Sweden, the costs of occupational accidents and ill health attributable to work-related stress are estimated to be above 4% of the GDP (€ 136.71 per worker) based on healthcare costs, sickness absence costs, and loss of productivity, etc. [13]. As the prevalence of mental illnesses increases, stress and depression are anticipated to be the primary causes of lost disability-adjusted life year by 2030. As a result, innovative and cost-effective solutions for this challenging occupational health issue are increasingly sought.

The types of interventions that have been studied have emphasized individual-centered interventions (i.e., problem solving) [14,15,16], adequacy of the interventions delivered [17,18,19], and interventions where the worker and supervisor identify challenges and develop a plan for new working arrangement (i.e., participatory interventions) [20]. Participatory work-focused interventions are recommended comprehensive strategies for addressing the causes of work stress and depression, such as psychosocial and organizational hazards. It has been suggested that an intervention designed to address work stress must be evidence-based, solving existing problems, planned, resourced, and implemented by those who are affected by stress or can be potentially affected [21]. The Productivity Measurement and Enhancement System or ProMes is an organizational, participative, “bottom-up” work-focused intervention that addresses work organization and environment, i.e., work-related risk factors, e.g., lack of autonomy and control, insufficient communication with co-workers, unclear and conflicting tasks, insufficient participation in decision-making, low esteem reward, and insufficient feedback [22,23]. Prior studies on the effectiveness of the widely studied intervention for productivity enhancement also suggest ProMes has preventive effects on the experience of stress, albeit with limited data [24,25]. Still, the link between stress reduction and actual business performance as an outcome measure is absent for most organizational studies.

Furthermore, few studies have conducted an economic evaluation of participatory work-focused organizational interventions [26]. In these identified studies, objective work-related outcomes have not been used to investigate the cost of the intervention in relation to its benefits. The effectiveness of previous interventions has been evaluated based on outcomes such as duration and recurrence of sickness absence, quality-adjusted life years (QALYs), and other health status or symptoms. The studies mostly identify one major cost, that is, lost work productivity associated with work-related stress. At times referred to as employer cost from the company perspective, this cost takes the form of work absences, presenteeism, and work sickness absences. Some studies show the cost of lost productivity due to presenteeism can even exceed that of work absences and work sickness absences. This might be attributed to the prevalent situation where employees who have stressful workplaces continue to go to work even when their health suffers (i.e., presenteeism) might be quite common [27,28]. Employees who are not able to tolerate the high levels of stress, on the other hand, might also risk a longer period of sick leave after choosing sick leave [8]. 

The common procedure of cost-effectiveness analysis is to compare more than one strategy using incremental cost-effectiveness ratios (ICERs). This is often the results of the economic analysis of a randomized controlled trial (RCT) regarded as the standard method for evaluating healthcare interventions [29]. RCTs, on the other hand, are infeasible or impractical to utilize for many organizational interventions. Policymakers may desire to analyze the influence of certain policies, such as the dissemination of recommendations on work practices activities, such as health campaigns, within local job contexts. As a result, the use of quasi-experimental designs is gaining popularity. This implies that for such organizational interventions, studies may not necessarily have a comparison strategy; thus, it is not possible to calculate ICERs. A before and after study design is a frequent quasi-experimental methodology used to examine such treatments. That is, before the intervention, some measure of compliance or result is taken, and then after the intervention, the same measure is taken.

The goal of this study was to evaluate the cost-effectiveness of an organizational intervention, the Productivity Measurement and Enhancement System, or ProMes, to minimize employee work-related stress and enhance work performance. Our paper addresses the question relating to the cost of the ProMes intervention and benefits in terms of objective workload measures among primary healthcare employees at risk of stress-related illness. In the present study, we argue that co-creation in the design and development of the intervention by engaging teams (the participants) will support the subsequent realization of reduction in work stress and business results in terms of work performance.

## 2. Materials and Methods

### 2.1. Study Design 

The study builds further on the Stress Prevention at Work (SPA) study, thoroughly described in two earlier papers [24,30]. In one of these papers, the primary outcome was job strain. Secondary outcomes were effort–reward imbalance, exhaustion, sleep, and recovery. However, for the present study, we collected detailed data about the costs of the intervention in a primary healthcare unit. Furthermore, we have data on objective work performance and stress both before and during the study. The study design was a prospective cohort with before and after measures of objective work performance and costs. Data on stress used in this study were collected by a single item stress question, described under Measures Section 2.3. The question was included in two questionnaires, one two months before the intervention started (May 2013) and one in the other half of September 2013 (the intervention itself started with a whole day workshop at the end of September). The same single item stress question was then distributed by weekly text messages [31] during two time series: SMS-series 1 (12 weeks) starting 4 October 2013, ending in December 2013 and SMS-series 2 (26 weeks) starting six months after the second baseline (March 2014) and ending in September 2014. The primary care unit was organized and applied new public management strategies in their everyday work routine with electronic systems supporting the reporting of performance. Thus, for objective performance data, the primary care units’ register data were used.

### 2.2. Study Participants

All employees of the units (i.e., a primary healthcare unit in one Swedish county council) who were employed at the time for the Stress Prevention (SPA) study were included in the study. Staff who worked on an ad hoc basis (i.e., hourly employees) were excluded from the study. There was some change in the number of employees at each measurement (*n* = 57 at baseline, *n* = 59 at 6 months follow up, and *n* = 67 at 12 months follow up) due to new recruitments, parental leave, staff attrition, and other factors. At baseline, 49 (86 percent) of respondents responded, followed by 50 (85 percent) at 6 months, and 55 (92 percent) at 12 months. At the start of the study, the average age of the participants was 44 years, with 42 (86 percent) being female and 39 (80 percent) having a university education. According to the statistics on professions in Sweden, the percentage of female employees tends to dominate in the various professions of the healthcare sector. This implies that our sample was representative of the percentage of the female population of healthcare employees in Sweden [32]. Nurses (*n* = 13, 27 percent), physiotherapists (*n* = 7, 15 percent), physicians (*n* = 10, 21 percent), and medical secretaries (*n* = 6, 12 percent) were the most common professions. Midwives, laboratory technicians, assistant nurses, and counselors were among the other occupations represented, as shown in Table 1.

### 2.3. Measures

#### 2.3.1. Objective Organizational Measures of Quantitative Workload

In the Swedish healthcare sector, productivity is typically assessed in terms of visits, surgical procedures, number of patients, and other factors [33]. As a measure of objective work performance, we used the ratio between the time unit (working hours) and the applicable productivity output used at the studied primary healthcare unit. This productivity ratio, such as the resource-to-results ratio, allows comparison of workload over time (and even between units), just as the resource-to-results ratio allows comparison of efficiency [33].

Data about objective work performance (hours worked, number of tasks, number of patient visits, number of administrative tasks, and number of phone calls answered) were collected every month from the central administration office of the county council. These data were collected from each healthcare unit separately. This means that every separate unit´s objective data in the SPA study were related to their employees’ stress experience, and in that way, the type of unit was controlled for. The data were used in the analysis computed into four quantitative workload ratios (Table 2). These were the ratios between the monthly total number of working hours as a nominator and the following denominators: (a) number of tasks, (b) number of patient visits, (c) number of administrative tasks, and (d) number of phone calls answered. The “number of tasks” is the organization’s own measure of the total number of monthly tasks, consisting of all their tasks, i.e., not only the sum of phone calls, number of visits, and number of administrative tasks.

#### 2.3.2. Stress

A weekly text message system, SMS Track^®^ (SMS-Track ApS, Esbjerg, Denmark) [31], was used to administer a validated single-item stress question (SISQ) from QPSNordic34+ [34]. “Stress means a state in which a person feels tense, restless, nervous or anxious or is unable to sleep at night because their mind is troubled all the time. Do you feel this kind of stress these days?” [34,35,36]. On a 5-point Likert scale, responses ranged from (1) “not at all” to (5) “very much”. Every Friday, automatic SMS messages were issued, and non-responders received an automated reminder the following Sunday. The weekly stress data were summarized to get monthly values that were used in the analysis (Table 2).

#### 2.3.3. Costs of the ProMes Intervention

We estimated the costs of the ProMes intervention in terms of direct (i.e., training, intervention design, and administration) and indirect (i.e., lost opportunities for alternative activities) expenses. The total costs of the ProMes intervention were estimated by summing the direct and indirect costs (Table 3). We adjusted all monetary amounts to 2020 values using the Consumer Price Index (CPI; Statistics Sweden, 2020) to account for inflation. Monetary values were expressed in Swedish Crowns, with 0.098 Euros equaling one Swedish Crown.

Direct costs of the ProMes intervention were associated with initial training, goal setting, and ongoing quality improvement activities, including (a) organizational readiness and management approval; (b) initial orientation and senior leader training; (c) ProMes development, including design team sessions for goal setting among employees and, when applicable, in collaboration with an intervention facilitator; (d) data collection and management; (e) administrative coordination, and (f) materials and other logistic supplies. These activities generated costs through trainer fees, transportation expenses, staff salary and benefits, and purchase of materials. All expenses regarding intervention activities were summed to produce a total direct cost of ProMes.

Indirect costs of the employees who participated in implementation-related activities also experienced indirect (i.e., “opportunity”) costs because of lost time spent on usual professional activities. These costs can be estimated in different ways by using the value of alternative uses of time or expenses on a person’s time (i.e., nominal wage) [37]. In this study, we used the value of nominal wages employees’ time uses. We separately estimated indirect costs to (a) health professionals, (b) clinical supervisors, (c) senior administrative staff for their participation which contributed to the costs of the ProMes intervention. To estimate the total indirect cost of ProMes, we first summed the time requirements for all relevant activities, including all in-person activities, such as regular participation in design teams. Next, we multiplied each estimate of lost productivity by the respective hourly wage for a given role. Then, we summed the products, for each professional role, of (a) the average estimated indirect cost and (b) the average number of individuals per design team per profession.

Total costs were calculated when direct and indirect costs were summed. The total cost of time resources by professional categories was estimated by multiplying the number of people assigned to a task by the total time to participate to obtain the total person-hours per task. Thereafter, the total of person-hours estimated by the professional category was multiplied by the hourly cost of the professional category.

### 2.4. Statistical Analysis

The costs and results associated with implementing ProMes were examined using cost-effectiveness analysis (see Tompa et al., 2010) [37]. We conducted the analysis from a business perspective for a healthcare organization because the organization (1) incurred direct and indirect costs because of participating in the intervention, and (2) reaped clinical benefits, such as stress reduction and improved objective work performance, because of participation. The analysis was divided into two parts: (1) estimating the change in stress and objective performance outcome, and (2) calculating an average cost-effectiveness ratio (ACER), i.e., the ratio of the cost to the benefit of an intervention without reference to a comparator.

Two effect sizes were estimated. First, we used baseline and follow-up measures to assess changes in the outcome variables by calculating mean effect sizes before–after intervention from an independent *t*-test. Objective performance data were measured repeatedly over a 17-month period, as described in the study design section. There were four data points before the intervention start in September (i.e., May–August 2013) and more data points after the intervention (From October 2013 until September 2014), and a clearly defined intervention start point, which meant that a time series analysis could be used. 

Second, the analysis used a time series regression (ITS) by centering the data around the time point of interest, when the intervention started in September 2013, to evaluate the changes that the intervention had on the outcome objective performance and stress. The Durbin–Watson statistic was used to test for first-order autocorrelations by performing a linear regression with the specification of the outcomes as the dependent and timepoints as the independent variables [38].

To combine the impacts and costs of the intervention, we first calculated the effect size of Cohen’s *d* to express changes in objective workload (i.e., performance) and stress levels (i.e., the standardized mean difference between after and after measures) [39,40]. Second, a change in the slopes of the regression lines was estimated (calculated as postintervention minus preintervention slope). An average r ratio (ACER) was calculated as a ratio of the average cost of the intervention and the effect sizes of the different outcomes to reflect the average cost per clinician for each unit change in outcome.

## 3. Results

In the first step, we analyzed the data in relation to changes before and after intervention start, in terms of the difference in average scores and effect sizes (Cohen’s *d*) from an independent *t*-test. There were no missing values for the objective workload metric. The results showed a slightly higher overall task/time ratio at pre-intervention (before and after a mean difference of 0.0487), implying that more tasks were done per hour worked pre-intervention (Table 4). The ratios for visits and administration were higher at post- than pre-intervention, indicating that there was a decrease in efficiency with more hours worked per visit completed/administrative tasks done (the higher ratio is negative). Moreover, the results showed increased efficiency in telephone calls taken per hour worked after the intervention compared to before the intervention. Concerning the stress outcome, the difference indicates higher stress levels after the intervention compared to before the intervention.

As a next step, we performed the time series analysis to investigate the change in trend defined as the difference between post- and pre-intervention slopes. In Table 5, the time series model showed that the observations for visits per hour worked, administrative tasks per hour worked were increasing before the intervention, but tasks per hour worked, stress levels, and calls taken per hour worked were decreasing. None of the outcomes had statistically significant preintervention trends. However, when looking at the difference between post- and pre-intervention slopes, the results showed a continuing decrease in stress levels, calls taken per hour worked. Visits per hour worked and administrative tasks per hour worked had also decreased but tasks per hour worked showed an increase. The difference between post- and pre-intervention slopes was small and showed a nonsignificant change in trend.

The effect size calculated from the change in the slopes of the regression lines was calculated (as post-intervention minus pre-intervention slope), which showed a slightly increased but nonsignificant efficiency in task/time ratio (post and pre mean difference of 0.029), implying that more tasks were done per hour worked (Table 5). The ratio for administrative tasks per hour worked showed rather an overall decrease, implying that more administrative tasks per hour worked. It was similar for the number of patient visits per hour worked, which had decreased post-intervention, indicating more visits completed per hour worked. The results showed no change in effect for telephone calls taken per hour worked and stress levels.

For standard ITS analysis, when missing or having an incomplete outcome, it may present potential problems since the time series data have temporal properties. In the analysis, some methods were used in the imputation of data for the stress outcome using mean values and linear interpolation methods (Table 5). No changes in the findings occurred for the time series specific method, i.e., linear interpolation. The time series model estimates implied that the ProMes intervention had a short-term change in the objective workload and stress in the quarter immediately after the intervention start. See also Appendix A, Figure A1 for the change in the trend of monthly values used in the model and Figure A2 for the predicted values from the model.

### Cost-Effectiveness

Each outcome measure’s average costs, effect sizes from the independent *t*-test and ITS, and ACERs are listed in Table 6. Based on a per-clinician cost of EUR 1742 and an effect size (*d* = −0.425, *t*-test), an expenditure of EUR 4099 per clinician was linked with each unit’s rise in stress levels, according to the ACER for stress. In addition, the expenditures per clinician associated with each unit change for various tasks where efficiency decreased was EUR 3062 for overall tasks per hour worked, EUR 924 for visits per hour worked, EUR 1080 for administrative tasks, but EUR 5732 for increased efficiency in answering phone calls. Based on the results of the ITS analysis, an expenditure of EUR 41,487 was linked with no change in stress levels, according to the ACER for stress. Furthermore, the expenditures associated with each unit change were EUR 3319 for overall tasks per hour worked, EUR 2761 for visits per hour worked, EUR 2880 for administrative tasks, but EUR 9123 for answering phone calls.

## 4. Discussion

This study evaluated the cost-effectiveness of the ProMes intervention in relation to the employees’ experience of work-related stress (measured by weekly text messages) and in relation to work performance (measured by organizational objective workload data). The study used a before and after design, independent *t*-tests, and an interrupted time series (ITS) analysis to examine whether the intervention had a substantial influence on work-related stress and objective workload. According to the independent *t*-test, our findings indicated that ProMes is not a cost–effective intervention since stress levels increased and efficiency (in terms of, for example, visits and administrative tasks completed per hour worked) decreased during the intervention, at extra cost to the primary healthcare unit. However, there was an increased efficiency in telephone calls taken per hour worked after the intervention compared to before the intervention. Phone availability has been one of the prioritized improvement areas during the primary healthcare units’ work with ProMES. According to the analysis, this was done at a minimal extra cost. 

Our findings showed that the results from the cost-effectiveness analyses using stress and objective workload outcomes were less favorable for ProMes. Compared with a few published syntheses on economic evaluations of occupational health interventions focused on the mental health (i.e., stress and depression) of working populations, our findings had a negative economic outlook. It must be made clear that these studies mostly conducted their analysis from a different perspective, such as the broader societal perspective, or used a different study approach in the evaluation of work-stress interventions [41,42,43]. Nonetheless, the findings give important information for occupational health and safety decision-makers. These findings may aid in the prioritization of preventive efforts in the setting of limited preventative resources, as well as the selection of the best choice among the numerous types of expenditures, which are often at the discretion of employers.

The effect sizes calculated from the change in the slopes of the regression lines were estimated (calculated as post-intervention minus pre-intervention slope) showed a completely different picture with increased efficiency in tasks performed per hour worked, administrative tasks per hour worked, as well as the number of patient visits per hour worked. For telephone calls taken per hour worked and stress levels, the results did not show any change in effect. According to the ITS, our findings indicated that ProMes is not a cost-effective intervention since it did not show any effect on stress levels and telephone calls taken per hour worked. For increased efficiency (in terms of, for example, tasks performed per hour worked, administrative tasks completed per hour worked, and visits per hour worked), ProMes might be cost-effective at a relatively high extra cost to the primary healthcare unit. Like the independent *t*-test results, none of the effect sizes from the ITS showed statistically significant results. What makes this method of analysis important is that it is possible that the ITS method revealed some potential biases in the estimate of the effect of the ProMes intervention that were incorrectly attributed to the observed effect, such as increased efficiency in telephone calls taken per hour worked. An advantage of an ITS design is that it allowed for the statistical investigation of, for instance, the secular trend bias (i.e., the outcome may be increasing before the intervention) in a before-and-after study design. 

To further comment on the economic evaluation, a subjective threshold (such as λ), cost-effectiveness thresholds (CETs) are typically used in determining whether an intervention under consideration is cost-effective or not compared to an ICER. If an ICER is less than the CET, the intervention is considered cost-effective; if it is greater than the threshold, the intervention or program is considered ineffective. Surprisingly for cost-effectiveness analysis, a wide range of thresholds are suggested at various levels of comparison: internationally, nationally, and by methodological approach. See, for instance, the debate about the gross domestic product (GDP) based CET at the national level and other methods [44,45]. Nevertheless, there is no consensus on how thresholds should be determined, especially for interventions from the organizational perspective. Previous studies have dealt with such concerns by referencing previous work that has suggested standards for thresholds or by presenting their results using a variety of thresholds [46].

To suggest whether an intervention under consideration is cost-effective or not compared to the ICER from the organizational perspective, previous studies have indicated the Cost–benefit analysis (CBA) might be most salient since it can provide a direct assessment of the impact on the bottom line [47]. This method assumes that the organization is driven by financial outcomes, as such outcomes are converted to money terms. This approach is useful and relevant if the company is interested in monetary outcomes for industrial and public relations. If the company is interested in nonmonetary outcomes, and a CEA is performed, the net benefit framework can also be used to derive financial outcomes for decision-making. The decision rule then becomes, if the net benefit is negative, the intervention is considered not cost-effective; if it is positive, the intervention or program is considered ineffective. In this study, where the effect size was used, we still arrived at a converted monetary term to obtain an ACER by dividing the average cost of the intervention by the effect size (which can be no effect, small, medium, or large effect). Since an effect size is also a standardized measure to compare different outcomes, the good practice of reporting the consequences can be followed, particularly in cases where important outcomes (e.g., health, worker morale, job satisfaction, health perceptions, product/ service quality, and customer relations are difficult to monetize) are used. In this study, the outcome measures were quantitative in nature, lacking measures on the qualitative aspect of work. Research on the aspects of qualitative work performance will contribute positively to occupation stress management if these measurement challenges are overcome. The lack of attention to qualitative work performance highlights the old saying that “what gets attention gets managed—especially if it is measured”. Indeed, in the economic evaluation sense, ‘‘what gets measured gets managed’’ since health and wellbeing providers, especially those involved in the purchasing decisions of occupational health and safety interventions, would like to realize quantifiable investment returns that are attractive.

Economic evaluations are often conducted in the context of incomplete information and uncertainty, which necessitates the use of proxy measures, and invariably, the need to make assumptions about the methods and unit prices used for valuing resource use, the methods used for dealing with incomplete data. Therefore, sensitivity analyses should be undertaken to assess how study results would change for different key assumptions and parameter values (i.e., the robustness of study results). We conducted sensitivity analyses to examine how CERs were influenced by variations in two key parameters: (1) effect sizes and (2) dealing with incomplete data. Alternative CERs were calculated using the effect sizes’ minimum and maximum values for each outcome (See Table A1 in Appendix B). The study’s conclusions remained unchanged. In their sensitivity analysis, Dopp [46] estimated alternative CERs for the lowest and maximum possible values for parameters, such as utilizing hospital reimbursement rates instead of staff hourly wage. For standard ITS analysis, potential problems could arise when outcome data are missing or incomplete. Since the time-series data has temporal properties, only some of the statistical methodologies may be appropriate. In this study, we used two approaches to impute the missing information for stress outcome: (1) non-time series-specific method, i.e., mean imputation, which is appropriate for stationary time series, assuming no trend or seasonality, and (2) time series-specific method, i.e., linear interpolation, which assumes that adjacent observations are like one another [48]. Yet again, the study’s conclusions remained unchanged.

Evidently, some dramatic changes have occurred in the work environment and working arrangements of healthcare workers since the study was conducted. Today, most people perform work remotely from outside the traditional offices of their employers. The situation has been somewhat different for a few employees in the healthcare sectors, but for many of them in the profession, they have experienced a historic record number of deaths but at the same time had infected COVID-19 patients to attend to. With the significant changes in their daily routines and workload due to COVID-19, the mental health of healthcare workers is at risk because of the high levels of stress. In such stressful situations, maintaining satisfactory levels of employee performance in terms of the quality of healthcare services and patient safety can be challenging as well as undermined [49,50]. Therefore, a health and safety culture is needed, especially mental health interventions at the individual, group, and organizational level among healthcare workers. The interventions need to address the impact of COVID-19 on the mental health of healthcare workers by identifying the job demands and resources and understanding the barriers to achieving effective outcomes in terms of health and well-being within their work environment [51].

### Strengths and Limitations

The study had several strengths, including the use of real (rather than estimated) implementation costs for ProMes and the longitudinal monitoring of outcomes (i.e., stress at work and work performance). Moreover, as noted previously, we interpreted the results of the present cost-effectiveness analysis by using the CERs, which were calculated from effect sizes to determine whether an intervention has a greater-than-zero impact and, if so, how large that effect is. The following are some of the reasons why effect sizes are useful in that regard: (1) it is a standardized metric that can be compared across research regardless of the scale used to assess the dependent variable, and (2) it is a standardized metric that can be compared across studies regardless of the scale used to evaluate the dependent variable [52]. There are theoretical and practical implications researchers want to be reminded of when using intervention effect sizes. It is important to remember that when calculating and evaluating effect sizes in economic assessments of organizational interventions, it might be vital to remember which effect sizes are appropriate for certain situations [52]. We selected approaches in this study that yield a generalizable effect size estimate by comparing pre- and post-measures. Nonetheless, the effect size (e.g., Cohen’s *d*) should be consistent regardless of the design utilized. 

Although the ICER may be more important to health economics and policy decisions, the ACER offers a number of advantages and practical qualities to consider: (1) it is a parameter that characterizes an intervention’s clinical and economic properties independent of its comparators (thus, application to one group or more than two groups is straightforward); (2) it conveys an intuitive meaning and interpretation (say, cost spent per year) that even laypeople can understand—it is very likely that researchers, policymakers, and payers will want to see the ACERs (e.g., for short vs. long-term costs) even when the ICER is appropriate for decision making. In most cases, a cost-effectiveness threshold, which is often required for ICERs, is not required. The incremental cost-effectiveness ratio, defined as the ratio of the difference in costs to the change in effectiveness between two competing strategies, has been widely recognized by academics and policymakers [37]. The average cost-effectiveness ratio (ACER) is the ratio of an intervention’s cost to benefit without considering a comparator. The ICER and ACER appear to estimate distinct parameters. Therefore, their objectives are different. Using ACERs, a decision rule based on a set budget may be devised to optimize total effectiveness, such as the idea of the budget for organizational interventions [53]. 

However, there are some limitations. First, this study did not have a control group, as would be the situation in a randomized design. Second, for such a complex intervention, a prolonged follow-up time of 18 months or more may have been appropriate. Researchers may want to find a balance between long follow-ups, implementation fatigue, and high attrition rates, especially if the implementation model requires service organizations to invest considerable resources in promoting employee health and performance. Third, in this study, no other indirect costs were used other than the opportunity cost considered in terms of time, person-hours. Opportunity costs increase the cost of undertaking workplace health promotion, and thus should be recovered whenever possible. Business owners might consider the opportunity costs whenever they decide about which of two possible actions to take. Future studies might want to include all opportunity costs when computing costs of work stress interventions to provide estimates for stakeholders, e.g., managers involved in purchasing such interventions to aid decision-making. Finally, researchers should also note that we did not consider the case with covariates in this paper. Covariates may be incorporated in the regression model or by other means. This suggestion, however, depends on the type of study design used. Interrupted time series (ITS) is a widely used quasi-experimental approach that evaluates the potential impact of an intervention over time. ITS may be used to address questions that are not feasible for a randomized design but with stronger assumptions. The pre-intervention trajectory is regarded as the control ‘period’ and the post-intervention trajectory as the intervention ‘period’ so that each individual acts as their own control. The difference between mean trajectories at the intervention time is then used to estimate the effect of the intervention [38]. 

## 5. Conclusions

In conclusion, using thorough organizational interventions, such as the ProMes in a primary care setting to reduce stress levels is not cost-effective. According to the ITS analysis, ProMes might be cost-effective for some objective performance measures but not statistically significant for increasing efficiency for tasks performed per hour worked, administrative tasks completed per hour worked, and visits per hour worked, albeit at a relatively high cost. Implementing ProMes, frequently necessitates significant modifications in organizational structure (for example, the continuation of interdisciplinary team meetings) and culture (e.g., increased emphasis on fidelity during supervision). As a result, implementation efforts inside clinical care organizations must be thoroughly examined to assure their effectiveness. Our research also offers an example of how to proceed for occupational health researchers looking to determine the cost-effectiveness of an intervention utilizing a design based on time series data without a comparator.

## Figures and Tables

**Table 1 ijerph-19-02431-t001:** Descriptive data, study participants (*n* = 49).

Variable	Description	
Sex, *n* (%)	Female	42 (86)
	Male	7 (14)
Age, years, mean (SD)	Mean age	44.4 (12.2)
Working hours, mean (SD)	Weekly working hours	37.3 (5.8)
Formal ed. level, *n* (%)	Secondary school	9 (18)
	University education	39 (80)
	Higher academic ed.	1 (2)
Profession, *n* (%)	Nurse	13 (27)
	Physiotherapist	7 (14.5)
	Physician	10 (20.5)
	Medical secretary	6 (12)
	Midwife	4 (8)
	Laboratory technician	3 (6)
	Assistant nurse	3 (6)
	Counselor	2 (4)
	Manager/Assist. Man.	1 (2)

**Table 2 ijerph-19-02431-t002:** Monthly mean stress and ratio of the total number of tasks (Tasks) to the total number of hours performed at the group level; Time and the total number of visits (home visits, group visits, and so on); Time spent on administrative tasks (Admin); total number of administrative tasks (Admin); Time spent on the phone and total number of answered phone calls (Phone).

Year/Month ¹	Tasks/Time ^2^	Time/Visits ^3^	Time/Admin ^3^	Time/Phone ^3^	Stress ^4^ (Mean/Month)
1305	1.57	1.43	2.08	1.86	2.9
1306	1.47	1.60	2.04	2.18	2.85
1307	1.74	1.41	1.76	1.64	2.8
1308	1.58	1.50	2.07	1.78	2.75
1309	1.39	1.67	2.39	2.04	2.7
1310	1.50	1.57	2.39	2.01	3.26
1311	1.41	1.63	2.28	2.14	3.21
1312	1.46	1.66	2.10	1.98	3.13
1401	1.48	1.69	2.15	1.87	3.11
1402	1.48	1.65	2.22	1.87	3.03
1403	1.49	1.69	2.25	1.78	2.96
1404	1.50	1.69	2.15	1.79	2.92
1405	1.53	1.69	2.10	1.76	2.86
1406	1.60	1.63	1.90	1.69	2.78
1407	1.64	1.58	1.90	1.61	2.67
1408	1.47	1.77	2.27	1.77	2.51
1409	1.44	1.69	2.26	1.92	2.59

¹ M0 was in May 2013 until intervention kickstarted in September 2013. Follow-up data were collected until September 2014. ^2^ should be interpreted as increased ratio = more tasks done per hour worked (increased efficiency). ^3^ should be interpreted as increased ratio = more hours worked per/number of tasks completed (decreased efficiency). ^4^ Stress is calculated as mean stress levels/month, thus interpreted as increases in mean value indicates increased stress levels.

**Table 3 ijerph-19-02431-t003:** Costs of the ProMES intervention (*n* = 49).

Cost Description	Total Cost (EUR)	Unit Cost (EUR)
Personnel time associated with intervention delivery (facilitators)	39,998	816
Preparation and planning of the intervention	3016	62
Training of staff, Kick-off	8623	176
Documentation of assessments	7667	156
Staff meetings concerning the intervention	839	17
Travel time, other supplies, and equipment	9541	195
Personnel cost of developing and implementing intervention	9631	197
Other miscellaneous staff-related time use	6065	124
Total	85,380	1742

Value of unit cost = per clinician. The sum of all the unit costs per clinician gives an average cost of EUR 1742.

**Table 4 ijerph-19-02431-t004:** Linear regression model, changes in outcome measures (before and after the intervention). A negative mean difference indicates increased efficiency/productivity.

Outcomes	Mean (Before)	Mean (After)	Mean Difference	95% CI	Effect Size	95% CI
Tasks/Time	1.550	1.501	0.049	(−0.047; 0.144)	0.569	(−0.489; 1.610)
Time/Visits	1.522	1.662	−0.141	(−0.224; −0.055)	−1.884	(−3.101; −0.623)
Time/Admin	2.068	2.164	−0.096	(−0.291; 0.099)	−0.560	(−1.613; 0.511)
Time/Phone	1.90	1.849	0.051	(−0.139; 0.241)	0.304	(−0.750; 1.348)
Stress ^1^	2.80	2.934	−0.134	(−0.490; 0.222)	−0.425	(−1.472; 0.636)

^1^ Negative mean difference in stress level indicates increased stress levels.

**Table 5 ijerph-19-02431-t005:** Interrupted time series analysis, change in trend post- and pre-intervention slopes.

Outcomes	Pre-Slope	Pre-Slope (*p*-Value)	Post-Slope	Slope Diff.	Slope Diff. (*p*-Value)
Tasks/Time	−0.023	0.440	0.006	0.029	0.337
Time/Visits	0.027	0.166	0.005	−0.022	0.259
Time/Admin	0.061	0.355	−0.014	−0.075	0.278
Time/Phone	−0.014	0.776	−0.031	−0.017	0.726
Stress	−0.051	0.672	−0.060	−0.009	0.943
Stress (mean imputation)	−0.054	0.565	−0.047	0.007	0.941
Stress (Linear interpolation)	−0.051	0.576	−0.059	−0.008	0.939

**Table 6 ijerph-19-02431-t006:** Average cost-effectiveness ratio for each outcome measure.

Outcomes	Cost (EUR)	Effect Sizes (*t*-test)	ACERs (EUR)	Effect Sizes (ITS)	ACERs (EUR)	Cohen’s *d* (ITS)	ACERs (EUR)
Stress	1742	−0.425	4099	−0.008	217,807	−0.042	41,487
Tasks/Time	1742	0.569	3062	0.029	60,085	0.525	3319
Time/Visits	1742	−1.884	925	−0.022	79,202	−0.631	2761
Time/Admin	1742	−1.613	1080	−0.075	23,233	−0.605	2880
Time/Phone	1742	0.304	5732	−0.017	102,497	−0.191	9123

## Data Availability

Data supporting the reported results in this study are reported in Table 1 and Table 2 presented in the Methods section, which makes the data already publicly available.

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
