# Peer review of "The Cost-Effectiveness Analysis of the Productivity Measurement and Enhancement System Intervention to Reduce Employee Work-Related Stress and Enhance Work Performance"

_ijerph, 2022, doi:10.3390/ijerph19042431_

Round 1

Reviewer 1 Report

I believe that the article deals with a very important topic of stress and productivity at work. I also have a few comments which I present below. Firstly I miss the source of information after this sentence in the introduction: It is the second most frequent cause of sickness absence in many European countries often after low back pain in EU countries such as the UK and Finland. The source in the next sentence is missing: In Europe, 38 the annual costs of absenteeism and lost productivity while at work (i.e., presenteeism) because of stress and depres-39 sion are estimated at € 72 billion. In the following parts of the article, there is also no reference to the literature in few cases.

There are also some linguistic issues in the article that could be improved, e.g. the word further is used twice in the first paragraph, maybe it is worth using a different word? I also have another linguistic comment, in this sentence: "Work-related stress impact significantly the health of individuals, organizations, and national economies". Does it mean that stress impact health of organizations and economies?

Additionally, sometimes the results of other studies are described in the present tense, and sometimes in the past tense. It should harmonized.

The study was conducted almost 7 years ago. What could have changed since then? How can we interpret the results now? How COVID-19 is related to stress at work? It is worth analyzing all these issues in retrospect. I believe that such a part should be added to the article.

I am also wondering what exactly were about the two previous articles based on this study? What are the main differences between them?

It is also worth considering why 86% of the respondents were women? Is this the typical percentage of women in this industry in Sweden? If not, what was the reason?

I am also thinking that the study only took into account the quantitative aspect of work, e.g. tasks / time, but it seems to me that in this type of work, the qualitative aspect is also very important, because a larger number of work in numbers may indicate a lower quality work ... Can this be included in the study in any way? Is the quantity or quality of work more important in this type of industry? Perhaps it is worth referring to this issue somewhere in the article.

In conclusion, I think that the article deals with a very interesting topic, but it should be improved. It is also important that the study was conducted almost 7 years ago, and since then a lot has changed when it comes to stress at work (e.g. COVID-19).

Author Response

I believe that the article deals with a very important topic of stress and productivity at work. I also have a few comments which I present below. Firstly I miss the source of information after this sentence in the introduction: After the It is the second most frequent cause of sickness absence in many European countries often after low back pain in EU countries such as the UK and Finland.

Thank you for pointing this out. We apologize for this mistake we do think the reference are very important. We have added references where appropriate and a sentence referring to absences attributed to work-related stress in the Swedish context.  

The source in the next sentence is missing: In Europe, 38 the annual costs of absenteeism and lost productivity while at work (i.e., presenteeism) because of stress and depression are estimated at € 72 billion. In the following parts of the article, there is also no reference to the literature in a few cases.

Again, thank you. We apologize for this mistake we do think the reference is very important. We have a few references that could have been placed at the right ends of the sentences. We have done so now.

EU-OSHA. Calculating the cost of work-related stress and psychosocial risks. European Agency for Safety and Health at Work, 2014

Matrix. Economic analysis of workplace mental health promotion and mental disorder prevention programs and of their potential contribution to EU health, social and economic policy objectives; Executive Agency for Health and Consumers Available at: http://www.mentalhealthpromotion.net/?i=portal.en.enmhp-news. 2900, 2013.

There are also some linguistic issues in the article that could be improved, e.g. the word further is used twice in the first paragraph, maybe it is worth using a different word? I also have another linguistic comment, in this sentence: "Work-related stress impact significantly the health of individuals, organizations, and national economies". Does it mean that stress impact health of organizations and economies?

The authors agree that this sentence is a bit confusing. We suggested this sentence instead. "Work-related stress has a significant impact on the health of individuals, as well as an impact on organizational and national economies." The authors hope that this sentence is a bit clearer to understand now.

Additionally, sometimes the results of other studies are described in the present tense, and sometimes in the past tense. It should be harmonized.

The authors noticed the disharmony in the manuscript. The authors have proofread the manuscript and we believe that manuscript has considerably improved.

The study was conducted almost 7 years ago. What could have changed since then? How can we interpret the results now? How COVID-19 is related to stress at work? It is worth analyzing all these issues in retrospect. I believe that such a part should be added to the article.

Please see line 361 for the suggested in relation to Covid-19.

Evidently, some dramatic changes have occurred in the work environment and working arrangements of health care workers since the study was conducted. Today, much of the work is performed remotely from outside of the traditional offices of the employers. The situation has been somewhat different for a few employees in the healthcare sectors, but for many of them in the profession, they have experienced a historic record number of deaths but at the same time infected COVID-19 patients to attend to. With significant changes in their daily routines and workload due to COVID-19, the mental health of health care workers has been shown to be at risk, which is associated with high levels of stress. In such stressful situations, maintaining satisfactory levels of employee performance in terms of the quality of health care services and patient safety can be challenging as well as undermined (Cheng 2020, Teoh 2020). Therefore, health and safety culture are needed, especially mental health interventions at the individual, group, and organizational level among health care workers. The interventions need to address the impact of COVID-19 on the mental health of health care workers by identifying the job demands, and resources and understanding the barriers to achieving effective outcomes in terms of health and well-being within a given work environment (Guisino et al., 2020).

I am also wondering what exactly were about the two previous articles based on this study? What are the main differences between them?

Good point! The authors believe that it is perhaps important to clarify even more than in this article we do not evaluate the effect of ProMes on stress in terms of job strain, for example, as was the case in the previous (main) article. This study is mainly about the cost-effectiveness in relation to productivity measures and stress as it is measured with SMS text messages. Therefore, the authors have changed the first sentence of the discussion ‘’This study evaluates the cost-effectiveness of the ProMes intervention, in relation to the employees’ experience of work-related stress (measured by weekly text messages), and in relation to work performance (measured by organizational productivity data).’’ The authors also try to clarify under the study design section the primary outcome used in the previous papers.

It is also worth considering why 86% of the respondents were women? Is this the typical percentage of women in this industry in Sweden? If not, what was the reason?

Thank you for your observation. According to the statistics on professions in Sweden, the percentage of female employees tend to dominate in the various professions of the healthcare sector. This implies that our sample is not different from the percentage for the population of health care employees in Sweden. The authors have added a reference showing that our sample is representative of the percentage of females, who tend to dominate in the healthcare sector i.e., women are overrepresented. Please see line 137 for the additional comment.

I am also thinking that the study only took into account the quantitative aspect of work, e.g. tasks/time, but it seems to me that in this type of work, the qualitative aspect is also very important, because the larger number of work in numbers may indicate a lower quality work ... Can this be included in the study in any way? Is the quantity or quality of work more important in this type of industry? Perhaps it is worth referring to this issue somewhere in the article.

This is an important point from the reviewer. In this study, the outcome measures are quantitative in nature, lacking measures on the qualitative aspect of work. We do not have such qualitative data from the organization that can be used in the article. Most medical staff, for example, usually complain about these measurement challenges. Please see line 339 for the authors' comments on qualitative work.

The authors have included in the discussion section to highlight the importance of collecting qualitative data in future studies. Research on the aspects of qualitative work performance will contribute positively to occupation stress management if measurement challenges are overcome. The lack of attention on qualitative work performance highlights the old saying that ‘‘what gets attention gets managed – especially if it is measured.’’ Indeed, in the economic evaluation sense, ‘‘what gets measured gets managed’’ since health and wellbeing providers, especially those involved in purchasing decisions of workplace health promotion interventions, would like to realize quantifiable investment returns that are attractive.

In conclusion, I think that the article deals with a very interesting topic, but it should be improved. It is also important that the study was conducted almost 7 years ago, and since then a lot has changed when it comes to stress at work (e.g. COVID-19).

Thank you for this suggestion. The authors think this is a good point. The authors have included in the discussion section a paragraph about COVID-19, stress at work, and work performance. Please see the suggested text in an earlier response on line 361.

Reviewer 2 Report

Dear Authors,

The article presents an interesting study on cost-effectiveness analysis of the ProMes intervention to reduce employee work-related stress and enhance work performance. The paper is generally well structured and meets the standards of scientific papers. The statistical analysis and results are clearly presented. The comments in the attached file are meant to enhance the quality of the presented study. Good luck in your future research!

Author Response

The article presents an interesting study on cost-effectiveness analysis of the ProMes intervention to reduce employee work-related stress and enhance work performance. The paper is generally well structured and meets the standards of scientific papers. The statistical analysis and results are clearly presented. The comments in the attached file are meant to enhance the quality of the presented study. Good luck in your future research!

The article presents an interesting study on cost-effectiveness analysis of the ProMes intervention to reduce employee work-related stress and enhance work performance. The paper is generally well structured and meets the standards of scientific papers. The statistical analysis and results are clearly presented.

The following comments are meant to improve the quality of the presented study:

  1. The Introduction requires some changes – it serves as theoretic introduction to the topic, but this part lacks depth. The theoretic deliberation shall be presented here in a more detailed way, describing the concept of work-related stress and work performance. Four paragraphs do not seem to exhaust the subject or even to present it with the appropriate standard of due diligence. Authors may present the current state of the research field. There is only one publication dating 2021 and eight dated 2020.

Thank you for this comment. As far as the authors know, the current systematic overviews of economic evaluation of interventions for mental health in the workplace have been included in the study. The introduction of the topic was limited to cover what the study aim is. The authors agree with the reviewer on describing the concept of work-related stress and work performance. The authors have included two paragraphs in the introduction section. Please see lines 39-60 for the suggested text.

  1. In the Materials and Methods part of the paper additional information is required:
  2. Study design – Authors should provide us with spatial extent of the study. Also, the time scope (2013 – 2014) raised doubts to what extent results and finding were still current and valid in 2022. Authors mentioned: “Data on stress was/ were collected via two questionnaires […]” – there is no further information about these research tools, or the scales used to measure the work-related stress and employees’ performance – were they created by the Authors? Were they adopted from studies conducted by other researchers? Please, specify the sources.

Thank you for comments. The authors have mixed up "questionnaires" in here. Data on stress used in this study was collected by a single item stress question, described under Measures section 2.3.2. with the source of the question. The authors view on the time scope of the study is relevant since work-related stress is still as relevant for employee’s health and work. This is an interesting topic and data looks at another dimension concerning economic outcomes. In our earlier published articles on the data, the study had not focused on economic evaluation at all, and that is why this information is still useful to add to the literature.

I have written suggestions for changed wording under study design

  1. Study participants – detailed information about study participants is required. Authors may organise the study participant characteristics in a form of a table for better legibility.

Thank you for this comment. The authors have included a table 1. as suggested. Please see line 145.

  1. Measures (2.3.3. Costs of ProMes intervention) – were there any indirect costs focused on other than the opportunity cost (line150)?

Thank you for drawing the authors attention to this. In this study, no other indirect costs were used other than the opportunity cost considered in terms of time, person-hours. The authors have mentioned this exclusion in the limitation section.

  1. The Discussion part– Authors discussed their results, but some adjustments are suggested:
  2. comments are necessary on how they can be interpreted in perspective of previous studies/ other researchers’ findings, existing theories. In the Discussion, the Authors could stress on how they enrich the existing theories, what is novel in their studies and why it is important. Please, elaborate on the theoretical and practical implications of the study.

Thank you for this point. We have commented on the findings in our study in relation to previous studies/ other researchers’ findings, existing theories. However, the authors feel that we have somewhat covered the theoretical and practical implications in the discussion of the study. The authors think that we show the importance of the theories and methods used in the study. In fact, the study is just about how existing theories and methods can be applied to certain study design by conducting an economic evaluation.

  1. Also, limitations may be clearly addressed, and future research directions are missing.

Thank you for mentioning. The authors have added comments to clearly address some of the study’s limitations.

  1. Other comments:
  2. Are the study findings relevant to the to the current Covid-19 situation? Could the Authors comment on how global pandemics influenced the employee work-related stress and performance?

Thank you for your suggestion. The authors have inserted in the discussion section a paragraph touching on the topic COVID-19, stress at work and work performance. Please see the suggested text on line 361 in relation to Covid-19.

‘’Evidently, some dramatic changes have occurred in the work environment and working arrangements of health care workers since the study was conducted. Today, much of the work is performed remotely from outside of the traditional offices of the employers. The situation has been somewhat different for a few employees in the healthcare sectors, but for many of them in the profession, they have experienced a historic record number of deaths but at the same time infected COVID-19 patients to attend to. With significant changes in their daily routines and workload due to COVID-19, the mental health of health care workers has been shown to be at risk, which is associated with high levels of stress. In such stressful situations, maintaining satisfactory levels of employee performance in terms of the quality of health care services and patient safety can be challenging as well as undermined (Cheng 2020, Teoh 2020). Therefore, health and safety culture are needed, especially mental health interventions at the individual, group, and organizational level among health care workers. The interventions need to address the impact of COVID-19 on the mental health of health care workers by identifying the job demands, and resources and understanding the barriers to achieving effective outcomes in terms of health and well-being within a given work environment (Guisino et al., 2020).’’